# A 24-to-30 GHz Ultra-High-Linearity Down-Conversion Mixer for 5G Applications Using a New Linearization Method

**DOI:** 10.3390/s22103802

**Published:** 2022-05-17

**Authors:** Shenghui Yang, Kejie Hu, Haipeng Fu, Kaixue Ma, Min Lu

**Affiliations:** 1School of Microelectronics, Tianjin University, Tianjin 300072, China; yangshenghui@tju.edu.cn (S.Y.); hkj@tju.edu.cn (K.H.); makaixue@tju.edu.cn (K.M.); 2State Key Laboratory of Mobile Network and Mobile Multimedia Technology, ZTE Corporation, Shenzhen 518000, China; lu.min1@zte.com.cn

**Keywords:** 5G, SiGe BiCMOS, linearization techniques, millimeter waves, mixers

## Abstract

The linearity of active mixers is usually determined by the input transistors, and many works have been proposed to improve it by modified input stages at the cost of a more complex structure or more power consumption. A new linearization method of active mixers is proposed in this paper; the input 1 dB compression point (IP1dB) and output 1 dB compression point (OP1dB) are greatly improved by exploiting the “reverse uplift” phenomenon. Compared with other linearization methods, the proposed one is simpler, more efficient, and sacrifices less conversion gain. Using this method, an ultra-high-linearity double-balanced down-conversion mixer with wide IF bandwidth is designed and fabricated in a 130 nm SiGe BiCMOS process. The proposed mixer includes a Gilbert-cell, a pair of phase-adjusting inductors, and a Marchand-balun-based output network. Under a 1.6 V supply voltage, the measurement results show that the mixer exhibits an excellent IP1dB of +7.2~+10.1 dBm, an average OP1dB of +5.4 dBm, which is the state-of-the-art linearity performance in mixers under a silicon-based process, whether active or passive. Moreover, a wide IF bandwidth of 8 GHz from 3 GHz to 11 GHz was achieved. The circuit consumes 19.8 mW and occupies 0.48 mm^2^, including all pads. The use of the "reverse uplift" allows us to implement high-linearity circuits more efficiently, which is helpful for the design of 5G high-speed communication transceivers.

## 1. Introduction

The fifth-generation (5G) wireless network is one of the most attractive research hotspots in recent years. To obtain wider bandwidth and higher communication rates, the frequency of 5G applications is increased gradually towards millimeter-wave bands. To save the time to market as much as possible, the circuits should cover multiple frequency bands to avoid repeated design when new applications appear, which brings greater design challenges to improve performance at the same time. Some broadband receivers covering multiple frequency bands have been proposed for 5G new radio (NR) frequency bands, including 24.5, 28, 37, 39, and 43 GHz [1,2].

In the receiver of a heterodyne structure, the down-conversion mixer is located between LNA and VGA, in the second stage. Parameters such as conversion gain (CG), 1dB compression point, noise figure (NF), and isolation cannot be ignored and seriously affect the system’s overall performance. Moreover, the data rate of several Gbps is highly expected, which requires the mixer’s intermediate frequency (IF) to be higher and have a wide IF bandwidth.

For passive mixers, resistive ring mixers [3] and drain-driven mixers [4] all have a significant loss at millimeter-wave frequencies, which requires additional power consumption to compensate, and higher local oscillator power requirements make the system more complicated. Various active implementations have been proposed under silicon-based processes, for example, gate-/base-driven [5], source-driven [6,7], bulk-driven [8], drain-driven and gate-driven [9], switching stage only [10], half Gilbert-cell [11], and switching stage with single-ended transconductance stage [12]. They all have strengths in terms of gain, linearity, bandwidth, and so on, but they cannot be improved simultaneously. The Gilbert-cell is widely used due to its double-balanced characteristics and moderate gain capability. Nevertheless, due to the limitation of its stacked structure and low voltage, its linearity is far inferior to passive mixers. The multiple-gate transistor (MGTR) method is applied to mixers [13,14,15], which improves the input third-order intercept point (IIP3) effectively without significantly increasing power consumption and sacrificing CG. However, it is voltage-sensitive, and the boosting effect is weakened rapidly when a large signal is input and the input-referred 1dB compression point (IP1dB) has not been raised. Source/emitter degradation is also a widely used linearization method [16], but the output-referred 1dB compression point (OP1dB) has not been improved effectively due to the expense of conversion gain. In [17], Gilbert-cell’s transconductance stage and switching stage are separated through a transformer; the linearity is improved through independent voltage bias. However, the introduction of large-area passive structures leads to higher costs and more complicated designs. Therefore, a more efficient method is needed to improve the linearity of the active mixers while sacrificing other performance less.

This paper proposes a new and more efficient linearization method to increase the IP1dB and OP1dB of Gilbert-cell-based active mixers by analyzing the phenomenon of “reverse uplift”. Compared to other works where linearity is concerned [4,6,16,18,19,20,21,22,23], no extra components are added except a pair of phase-adjusting inductors, the linearity is greatly improved by using the energy of harmonics, and the IP1dB reaches +7.2~+10.1 dBm. At the same time, the CG and the gain flatness remain appropriate, so the excellent OP1dB performance is realized simultaneously, with an average of +5.4 dBm, which is the state-of-the-art OP1dB performance in mixers under a silicon-based process, whether active or passive. Furthermore, the input radio frequency (RF) band covers the n257, n258, and n261 bands of 5G NR frequency bands. The use of a low-Q Marchand balun ensures that the IF covers 3~11 GHz and the relative IF bandwidth up to 114%.

This paper is organized as follows. Section 2 describes the principle of the proposed new linearization method. Section 3 presents a specific implementation of the high-linearity down-conversion active mixer. In Section 4, the measurement results are demonstrated; then, the conclusion is drawn in Section 5.

## 2. Principle of Linearity Improvement

### 2.1. Analysis of the Principle of “Reverse Uplift”

The Gilbert-cell-based structure is one of the most popular implementations of active mixers, consisting of a V/I conversion stage (converting voltage changes to current changes) and a switching core. The half Gilbert-cell is analyzed for simplification, as shown in Figure 1; we treat the transconductance stage as a memoryless non-linear system. Considering the harmonic distortion caused by non-linearity, the output signal can be approximately expressed as
(1)$$y\left(t\right)=\alpha_{1}x\left(t\right)+\alpha_{2}x^{2}\left(t\right)+\alpha_{3}x^{3}\left(t\right)+\ldots +\alpha_{n}x^{n}\left(t\right)$$
where $\alpha_{1}$, $\alpha_{2}$, $\alpha_{3}$, … $\alpha_{n}$ represent the coefficients of each order component, respectively. In general, when a cosine signal with an amplitude of M and an initial phase of zero is fed into such a system, a fundamental component $\left(\alpha_{1}M+\frac{3}{4}\alpha_{3}M^{3}\right)\cos\left(2\pi f_{in}t\right)\;$ and a second harmonic component $\frac{\alpha_{2}\cdot M^{2}}{2}\cos\left(4\pi f_{in}t\right)\;$ are generated due to nonlinear effects, where $f_{in}$ represents the signal frequency. The magnitude of the second harmonic component is proportional to $M^{2}$, the energy of higher-order harmonics is weak enough to have a significant effect. If the system is stable, the gain of the fundamental component will be compressed as M increases, so the condition of $\alpha_{1}\alpha_{3}<0$ needs to be satisfied.

However, the actual circuit system is not conducting unidirectionally; for example, the junction capacitors of the transistors will cause reverse feedthrough, which will bring a part of the signal back to the input with some attenuation and phase shift. Then, the extra components will be generated through the effect of frequency mixing. Parameters $k_{1}$ and $k_{2}$ are defined as the feedback factors of the fundamental component and the second harmonic component from the output to the input through some passive devices, respectively. The possible reversal caused by the phase change is also considered, so its range is
(2)$$-1<k_{1},\;k_{2}<1$$ Therefore, the total fundamental and second harmonic components at the input can be expressed as $Y_{1}\left(t\right)$ and $Y_{2}\left(t\right)$.
(3)$$Y_{1}\left(t\right)=k_{1}\left(\alpha_{1}M+\frac{3}{4}\alpha_{3}M^{3}\right)\cos\left(2\pi f_{in}t\right)+M\cos\left(2\pi f_{in}t\right)\;\;$$
(4)$$Y_{2}\left(t\right)=k_{2}\frac{\alpha_{2}M^{2}}{2}\cos\left(4\pi f_{in}t\right)\;$$
(5)$$Y_{mix}\left(t\right)=K\left(Y_{1}\left(t\right)\times Y_{2}\left(t\right)\right)$$
$Y_{1}\left(t\right)$ and $Y_{2}\left(t\right)$ will mix at the input, resulting in extra components as shown in Equation (5), where K represents the coefficient determined by specific circuits. The fundamental component in $Y_{mix}\left(t\right)$ can be expressed as
(6)$$K\left[\frac{k_{2}\alpha_{2}M^{2}}{2}\left(1+k_{1}\alpha_{1}\right)M^{3}+\frac{3}{8}k_{1}k_{2}\alpha_{2}\alpha_{3}M^{5}]\;\cos\left(2\pi f_{in}t\right)\;\right]$$ Additionally, the gain of the fundamental component can be updated as
(7)$$\begin{array}{cc} Gain=\alpha_{1}+[\frac{3}{4}\alpha_{3} & +\frac{k_{2}\alpha_{2}}{2}\left(1+k_{1}\alpha_{1}\right)]M^{2}\\ & +\frac{3}{8}k_{1}k_{2}\alpha_{2}\alpha_{3}M^{4},\;\left(-1<k_{1},\;k_{2}<1,\;M>0\right) \end{array}$$ Compared with $\alpha_{1}+\frac{3}{4}\alpha_{3}M^{2}$, the extra part in Equation (7) may increase the compression point to a certain extent. Where 1 dB compression should have occurred, the new gain component compensates for these losses, thereby increasing the 1 dB compression point. This possibility is analyzed below.

Let
(8)$$T_{1}=\alpha_{1}$$
(9)$$T_{2}=\frac{3}{4}\alpha_{3}+\frac{k_{2}\alpha_{2}}{2}\left(1+k_{1}\alpha_{1}\right),\;(-1<k_{1}<1)$$
(10)$$T_{3}=\frac{3}{8}k_{1}k_{2}\alpha_{2}\alpha_{3},\;\left(-1<k_{1},\;k_{2}<1\right)$$

As shown in Figure 2a,b, in order to compress the gain in the opposite direction as the input signal increases and make the system tend to be stable, it is necessary to satisfy $T_{1}T_{3}<0$. Furthermore, when $T_{2}T_{3}<0$ is satisfied at the same time, the curve of gain compression has a phenomenon of “reverse uplift”, which makes the compression point further increase. In other words, if the following conditions in Equations (11) and (12) can be satisfied, a new curve of gain compression can be generated artificially using this phenomenon, making the IP1dB “delay”.
(11)$$k_{1}k_{2}\alpha_{1}\alpha_{2}\alpha_{3}<0$$
(12)$$\frac{3\alpha_{3}}{\alpha_{1}}+\frac{2k_{2}\alpha_{2}}{\alpha_{1}}\left(1+k_{1}\alpha_{1}\right)>0$$

When the input is extremely small, the gain is still $T_{1}$, but the quadratic term $T_{2}M^{2}$ will take effect when the input power increases gradually and the curve enters to the "reverse uplift" area, leading to the increasing of IP1dB. After that, when the input power is large enough, the fourth-order term $T_{3}M^{4}$ begins to dominate and the gain will be compressed sharply.

### 2.2. Circuit Behaviors under “Reverse Uplift”

As for a Gilbert-cell-based active mixer, the gain compression point is mainly determined by the transconductance stage, which is used as a V/I converter (converting voltage changes to current changes) and directs the current into the switching core. Under this requirement, both the MOSFETs (metal-oxide-semiconductor field-effect transistors) and the BJTs (bipolar junction transistors) are biased in the amplification area so the input signal needs to be small, otherwise the working state of the transistors will be changed. However, when the problem of gain compression is analyzed, the effect of large signal will have to be considered. In this case, the input stage is no longer just a V/I converter, but also a gate/base pumped active mixer with a single transistor, as shown in Figure 3a,b. Therefore, when the input signal is strong enough, the fundamental signal component makes the transistor work in the switching state and becomes the LO (local oscillator) source of this mixing mode, as represented by Equation (3), and the second harmonic signal component fed back to the input at the same time becomes the small input signal, as expressed by Equation (4). At this time, if the fundamental component is strong enough to make the transistor act as an ideal switch, K in Equation (5) will be approximately equal to $\frac{2}{\pi }$.

In summary, in the case of small signal input, the Gilbert mixer’s operating mode is conventional, but in the case of considerable signal input, it is a superposition of the two operating modes, so the new curve of gain compression is also a superposition of the two curves, as shown in Figure 4. As the amplitude of the input signal increases, the mode of the single transistor gate/base pumped reaches the optimal local oscillator condition, and the second harmonic is mixed as a small signal to generate a new gain component. The existence of “reverse uplift” is predicted by Equation (7).

In order to generate “reverse uplift”, the key is mixing the fundamental with the second harmonic. Therefore, while amplifying the fundamental signal, it is necessary to simultaneously develop a strong second harmonic signal in the transconductance stage of the Gilbert-cell. As shown in Figure 5, where gm represents the small-signal transconductance of fundamental, $gm_{2}$ represents the small-signal transconductance of the second harmonic, and $gm_{3}$ represents the small-signal transconductance of the third harmonic. Regardless of the BJT or MOS transistor, $gm_{2}$ rises rapidly in the sub-threshold region due to the “exponential” characteristic and the “square law” characteristic, and a peak value exists, which should be selected for this design. At the same time, the value of $gm_{3}$ is close to zero, so its effect can be ignored reasonably.

### 2.3. Phase Condition and Amplitude Analysis of “Reverse Uplift”

From Equation (1) and its expansion, we can approximate that $\alpha_{1}$ characterizes the fundamental. However, $\alpha_{2}$ and $\alpha_{3}$ cannot be regarded as coefficients of the second and third harmonics simply, because they also characterize the DC and fundamental components at the same time. Therefore, for Equation (11), it is difficult for us to judge the positive and negative of each parameter independently, and it is not necessary. This condition is derived from $T_{1}T_{3}<0$, implying the presence of gain compression. Therefore, it is only necessary to ensure the stability of the amplifier during design, which can be carried out through simulation, so Equation (11) can be regarded as a known condition.

In addition, condition (12) can be transformed into a magnitude of “reverse uplift” greater than 0, which can be expressed as
(13)$$\begin{array}{cc} Gain_{max}-\alpha_{1}=T_{1} & -\frac{T_{2}{}^{2}}{4T_{3}}-\alpha_{1}\\ & =-\frac{{\left(\frac{3}{4}\alpha_{3}\right)}^{2}+\frac{k_{2}{}^{2}\alpha_{2}{}^{2}}{4}{\left(1+k_{1}\alpha_{1}\right)}^{2}+\frac{3}{4}k_{2}\alpha_{2}\alpha_{3}\left(1+k_{1}\alpha_{1}\right)}{\frac{3}{2}k_{1}k_{2}\alpha_{2}\alpha_{3}}>0 \end{array}$$

As mentioned above, in order to ensure that the amplifier generates the strongest second harmonic to ensure the emergence of the second operating mode, the base/gate voltage should be set in the sub-threshold region. At this time, the third-order transconductance value is small and close to 0, so the effect of third-order nonlinear coefficient $\alpha_{3}$ is ignored reasonably to simplify the analysis. Then, Equation (13) can be simplified to
(14)$$-\frac{\frac{k_{2}{}^{2}\alpha_{2}{}^{2}}{4}{\left(1+k_{1}\alpha_{1}\right)}^{2}}{\frac{3}{2}k_{1}k_{2}\alpha_{2}\alpha_{3}}>0$$

If $\alpha_{1}<0$, then from Equation (11) we know that $k_{1}k_{2}\alpha_{2}\alpha_{3}>0$, but this requires the numerator to be less than 0 to satisfy the condition, which cannot be established obviously. If $\alpha_{1}>0$, and then $k_{1}k_{2}\alpha_{2}\alpha_{3}<0$, the magnitude of the “reverse uplift” can be controlled by the value of the numerator, which is exactly what is needed. Note that the magnitude of “uplift” should be limited, otherwise the improving of linearity will be meaningless because the compression curve is too steep.

Take the common-source/common-base amplifier as an example; the output signal is opposite to the input signal generally, which means $\alpha_{1}<0$. However, under this condition, the “reverse uplift” cannot be achieved, which requires the introduction of additional phase control. As shown in Figure 6, in the half Gilbert-cell, we introduce a phase-adjusting inductor at the emitter of the transconductance stage. At 27 GHz, as the value of the inductor increases, the phase difference of fundamental between collector and base decreases from 127° to below 90°, which means that the output signal transitions from reverse to forward gradually. It should be added that the output phase has been shifted by 180° due to the effect of parasitic capacitors at high frequencies.

Due to the selection of the bias voltage, $\left|\alpha_{2}\right|$ inevitably increases, so in order to control the amplitude, the fundamental gain $\left|\alpha_{1}\right|$ should be reduced as much as possible and seek a trade-off between gain and linearity. The phase-adjusting inductor can achieve this requirement through the control of phase and gain. In addition, by controlling the input impedance of the switching stage of the Gilbert-cell, the fundamental gain can also be reduced properly. As shown in Figure 7a,b, with the increase in emitter length of the switching transistors, the real part of the input impedance of the switching core is greatly reduced, resulting in a decrease in gain. The imaginary part stays relatively small, which means that the impedance is no longer frequency-sensitive. Therefore, a large emitter length is needed for amplitude control.

### 2.4. Steps of Circuit Implementation

If the Gilbert-cell based structure is adopted for a high-IP1dB down-conversion mixer using the phenomenon of “reverse uplift”, the bias voltage of the transconductance stage should be mainly determined by $gm_{2}$. At the same time, gm should also be considered along with power budget to determine the size of the transconductance stage. By adjusting the transconductance stage amplification phase by inductors, as mentioned above, it should be determined whether the compression point curve of the conversion gain has a “reverse uplift”, which can be carried out by using the large-signal harmonic balanced simulation. Furthermore, the amplitude can be controlled by the size of the switching stage, because it determines the output impedance of the transconductance stage. Finally, parameter optimization, matching the network design for every port, passive design, and EM (electromagnetic) simulation are carried out iteratively. The whole design process is shown in Figure 8.

## 3. Implementation of a High-Linearity Down-Conversion Active Mixer

The mixer was manufactured using a 130 nm SiGe BiCMOS process based on Gilbert-cell. Figure 9a shows that the basic circuit is composed of bipolar junction transistors. Q1 performs the amplification of the fundamental RF signal and the generation of its harmonics, four Q2 transistors form a double-balanced switching core for frequency conversion, and a Marchand balun is used as the output load. At the same time, using phase-adjusting inductors L1, with proper choice of voltage bias and size of Q1/Q2, will make the “reverse uplift” possible. According to the previous analysis, the base voltage of Q1 is set to about 0.8 V to obtain stronger second harmonic energy. The emitter length of Q2 is 40 μm for low-input impedance, and the value of 80 pH is selected for L1 to meet the phase condition.

When the harmonics appear at node ①, where the second harmonic is stronger than normal due to the choice of bias voltage, the mixing of the fundamental and the second harmonic leads to additional fundamental and third harmonic components at node ②. Low-impedance load at node ③ and phase-adjusting inductors are used to meet the conditions for “reverse uplift”; finally, a better IP1dB is achieved. On the other hand, when the RF input signal is large enough, the mode of the Q1 could be regarded as a base-pumped mixer, not just with the principle of the Gilbert-cell mixer, which can also explain the "reverse uplift" in another way.

As shown in Figure 9b, like the way of source degeneration, with the increasing value of emitter inductors, the IP1dB increases continuously and the gain drops. However, in this work, before the 1 dB compression point, the gain has a proper rise to “delay” the gain compression. Compared with the traditional source-degenerating method, low-load impedance will not reduce the conversion gain significantly. The simulation results show that if the source degeneration is adopted, the exchange ratio of the gain and the IP1dB is almost 1:1. If the high linearity is to be achieved, gain will be greatly sacrificed. In contrast, this method improves IP1dB more and only sacrifices less gain. Therefore, OP1dB can be synchronously enhanced, which benefits the system. Compared with the MGTR method, the implementation is less sensitive to bias and can be used in engineering applications. Additionally, in addition to the phase-adjusting inductors, no additional circuits are added, which is easier to be implemented and is more efficient.

Figure 10 shows the bias circuits of the transconductance stage and the switching stage. The current mirrors are used to provide required bias voltage and current of the transistors. The values of some main components are listed in Table 1.

Transformer-based matching networks are used in the RF port and LO port for single-ended to differential conversion and perform impedance matching to 50 ohms simultaneously. This process provides two thick metal layers with a thickness of 3 μm and the detailed parameters of the passive components are shown in Figure 11a,b. The two coils use the M5 and M6 layers, respectively, while using the M1 layer as the ground. In order to obtain a wide IF bandwidth, the IF port uses a Marchand-balun-based matching network, as shown in Figure 12. R1 reduces the Q value of the matching network and makes the input impedance of the Marchand balun much closer to the output impedance of the switching core. R2 enables the circuit to maintain proper gain flatness in the entire IF band while matching to a 50-ohm load. The influence of R1 and R2 is shown in Figure 13.

## 4. Simulations and Measurements

The circuit was simulated by Advanced Design System (ADS) and Virtuoso System Design Platform. In addition, Ansys HFSS simulates all passive structures.

The chip was fabricated in 130 nm SiGe BiCMOS process, offered by STMicroelectronics. High -peed HBT (heterojunction bipolar transistor) in process library was adopted for Q1 and Q2 in Figure 9a, and Q3 in Figure 10. NMOS transistor in process library was adopted for M1 in Figure 10. The specific dimensions are shown in Table 1. The SiGe BiCMOS process provided four thin metal layers and two thick metal layers. All capacitors were MOM (metal–oxide–metal) capacitors, using metal layers to form finger structures. All resistors were poly resistors offered by process library. Other passive components, such as inductors, transformers, etc., were designed by thick metal layers, and simulated by Ansys HFSS. Figure 14 presents the microphotographs of the proposed high-linearity and wide IF bandwidth down-conversion active mixer’s chip with a size of 0.69 mm × 0.69 mm including all pads, using a 130 nm SiGe BiCMOS process. The DC pads were bonded out, VDD were 1.6 V, the total current were 12.4 mA, and the total DC power consumption were 19.8 mW. The measurement results show that the circuit is not sensitive to bias changes.

Except for the IF port, all measurements, including S-parameters, conversion gain, linearity, and isolation, were performed using on-chip probing. The instruments connection is shown in Figure 15a; the RF and LO ports are connected through GSG probes, and the IF output port is connected to the PCB board through bonding wires and then to the SMA interface through the transmission lines for testing, as shown in Figure 15b; the loss of transmission line has been calibrated. A four-channel vector network analyzer with a working frequency up to 67 GHz was used for the three-port measurement of the mixer, and a spectrum analyzer for noise measurement. A 100 kHz to 50 GHz pre-amplifier was used for linearity measurements to amplify the input RF signal because the design has an ultra-high IP1dB. In order to obtain general performance, a room temperature of 27 ℃ was set for measurement [24].

The simulation and measurement results of the reflection coefficient of the RF port are shown in Figure 16a. The result indicates that the S_11_ is less than −10 dB in the required operating frequency band of 24~30 GHz, and the bandwidth is much larger than needed. Similarly, Figure 16b shows the simulation and measurement results of the reflection coefficient of the LO port. In the range of 16~24 GHz, the S_22_ is less than −10 dB. Due to the application requirements of ultra-wideband for the IF port, a Marchand balun is used to achieve an IF range of 3~11 GHz; as shown in Figure 16c, in the entire IF band, the S_33_ is less than −5 dB. The results show that the return loss of each port is within an acceptable range, which meets the general system requirements. 

Figure 16d features the measured conversion gain versus local oscillator power at the LO frequency of 20 GHz, the RF frequency of 27 GHz, and the IF frequency of 7 GHz. The reason for choosing this group of frequencies is because they are in the center of the operating frequency band and are representative. When the LO power increases to a certain level, the CG no longer has an obvious enhancement, and the LO has reached the optimal driving state. In order to obtain the best conversion gain and linearity performance, we chose a fixed local oscillator power of +2 dBm to complete all measurements.

Table 2 illustrates the measured conversion gain versus RF frequency. For 5G communication systems, the LO frequency is often changed, rather than working at a fixed frequency. Therefore, within the design range, we simulated and measured the LO frequencies from 16 GHz to 24 GHz in steps of 1 GHz. When the LO frequency is 16 GHz, the IF band is 8~11 GHz, and the CG is −2.3 ± 1.5 dB; when the LO frequency is 17 GHz, the IF band is 7~11 GHz, and the CG is −2.5 ± 1.5 dB; when the LO frequency is 18 GHz, the IF band is 6~10 GHz, and the CG is −2.7 ± 1.6 dB; when the LO frequency is 19 GHz, the IF band is 5~10.5 GHz, and the CG is −2.7 ± 1.6 dB; when the LO frequency is 20 GHz, the IF band is 4~9.5 GHz, and the CG is −2.7 ± 1.6 dB; when the LO frequency is 21 GHz, the IF band is 3~8.5 GHz, and the CG is −2.8 ± 1.6 dB; when the frequency is 22 GHz, the IF band is 3~7.5 GHz, and the CG is −3.0 ± 1.5 dB; when the LO frequency is 23 GHz, the IF band is 3~6.5 GHz, and the CG is −2.9 ± 1.3 dB; when the LO frequency is 24 GHz, the IF band is 3~5.5 GHz, and the CG is −3.4 ± 0.9 dB. The results show that the mixers all show good CG performance under different LO frequencies without large loss, and the flatness is acceptable within a certain frequency range.

The effect of the linearity improvement is shown in Table 3. To demonstrate the effectiveness of this method, we selected different frequency combinations to simulate and measure the CG curves. Under nine different operating frequencies, the mixer’s simulation and measurement compression curves are displayed. As the input RF power increases, “reverse uplift” appears before gain compression, which aligns with the aforementioned theoretical analysis. 

The IP1dB simulation and measurement results are shown in Table 4. Under the condition that the LO frequency is from 16 to approximately 24 GHz and the step is 1 GHz, the results show the high linearity of the proposed mixer. The measurement results show that this new linearization method improved the IP1dB effectively. In the IF range of 3~11 GHz, an IP1dB of + 7.2~+10.7 dBm is achieved. At the same time, an ultra-high OP1dB is achieved, far exceeding other similar works under the silicon-based process because this linearization does not reduce the conversion gain significantly, and an average OP1dB can reach +5.4 dBm. The simulated curves of fundamental and third-order products are shown in Figure 17a, and the simulated IIP3 versus RF power is shown in Figure 17b. It can be understood in principle that this method directly improves IP1dB, but does not truly eliminate high-order nonlinear components, but just uses it to expand IP1dB, so the improvement of IIP3 is limited, which is also the focus of our follow-up work. Although IIP3 does not increase significantly with IP1dB, its value is reasonable and meets the application requirements. The simulated SSB (Single Side Band)-NF is 9.7~12.1 dB in different IF states. The measured RF to LO isolation is greater than 34.5 dB, which meets the system requirements. 

Table 5 shows the comparison results between this proposed mixer and other down-conversion mixers in similar frequency bands. The comparison results show that the mixer has excellent *IP*_1*dB*_ and *OP*_1*dB*_ performance, which is benefited from the use of the “uplift reverse” phenomenon. FOM and $FOM_{IF}$ are introduced to measure the overall performance, which are expressed as follows.
(15)$$FOM=10log_{10}\left(\;\frac{10^{\frac{CG\left[dB\right]}{20}}\times 10^{\frac{IP_{1dB}\left[dBm\right]}{10}}}{P_{DC}\left[W\right]\times 10^{\frac{P_{LO}\left[dBm\right]}{10}}}\right)$$
(16)$$FOM_{IF}=10log_{10}\left(\;\frac{10^{\frac{CG\left[dB\right]}{20}}\times 10^{\frac{IP_{1dB}\left[dBm\right]}{10}}\times IFBW\left[GHz\right]}{P_{DC}\left[W\right]\times 10^{\frac{P_{LO}\left[dBm\right]}{10}}}\right)$$

It should be noted that the *IP*_1*dB*_ in [6] and [19], and the *OP*_1*dB*_ in [4,16], and [20,21,22,23], are calculated by Equation (17). The *IP*_1*dB*_ in [4] is calculated by Equation (18). The *CG* in [18] refers to voltage conversion gain. Additionally, all data in the third column of Table 5 satisfy Equation (19).
(17)$$OP_{1dB}\left[dBm\right]=IP_{1dB}\left[dBm\right]+CG\left[dB\right]-1$$
(18)$$IP_{1dB}\left[dBm\right]=IIP_{3}\left[dBm\right]-9.6$$
(19)$$S_{11}<-10\;dB$$

For 5G communication systems, the improvement in the *IP*_1*dB*_ enhances the capability of anti-blocking. A high *IP*_1*dB*_ can effectively prevent the system from entering a saturated state when facing a large interference signal. At the same time, in the receivers, the realization of the high linearity mixer can reduce the design difficulty of other circuits, which is beneficial to the improvement of the overall performance.

The mixer achieves higher *IP*_1*dB*_ and *OP*_1*dB*_ than any other work in this frequency band, including many passive mixers, showing an excellent linearity performance. At the same time, it achieves an IF bandwidth of 8 GHz under a silicon-based process with a relative bandwidth of 114%. The overall layout is compact, and the power consumption is moderate. The FOM reaches 24.73, the $FOM_{IF}$ reaches 33.76; both show that the mixer has excellent overall performance.

## 5. Conclusions

In order to improve the linearity of mixers in the millimeter-wave band more efficiently, a new linearization method for the active mixer is proposed, and a down-conversion mixer with ultra-high IP1dB/OP1dB and wide IF bandwidth for the 5G NR band was designed. It was fabricated in a 130 nm SiGe BiCMOS process offered by STMicroelectronics. The use of the “reverse lift” improves the linearity of the proposed mixer greatly. The non-linearity of the transistors performs the linearization without adding any complicated structures, which is helpful for the low-cost design of 5G high-speed communication transceivers. As compared to other silicon-based publications in Table 5, the mixer demonstrates a measured IP1dB of +7.2~+10.1 dBm, an average OP1dB of +5.4 dBm, FOM of 24.73, and $FOM_{IF}$ of 33.76 with 8 GHz IF bandwidth, showing the best linearity performance among the published mixers. Furthermore, this method is based on the analysis of the fundamental amplifier; so, it is not just for mixer design, it also has excellent potential to be used in any amplifier-based circuit design with high linearity.

## Figures and Tables

**Figure 1 sensors-22-03802-f001:**
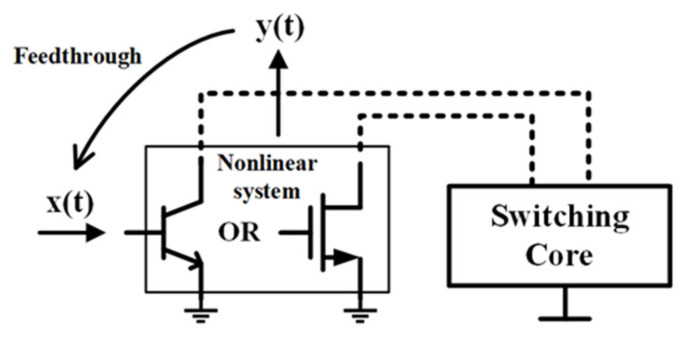
Block of nonlinear system.

**Figure 2 sensors-22-03802-f002:**
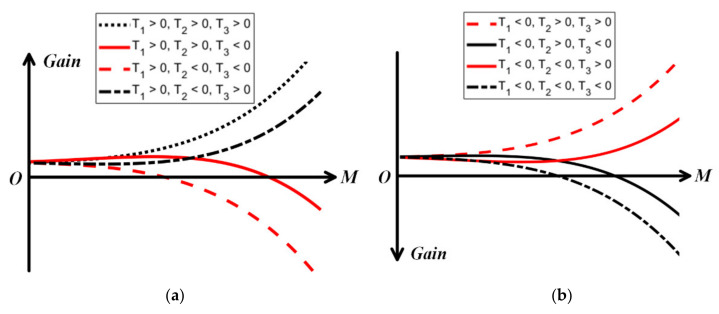
Different trends of Gain. (**a**) $T_{1}>0$, (**b**) $T_{1}<0$.

**Figure 3 sensors-22-03802-f003:**
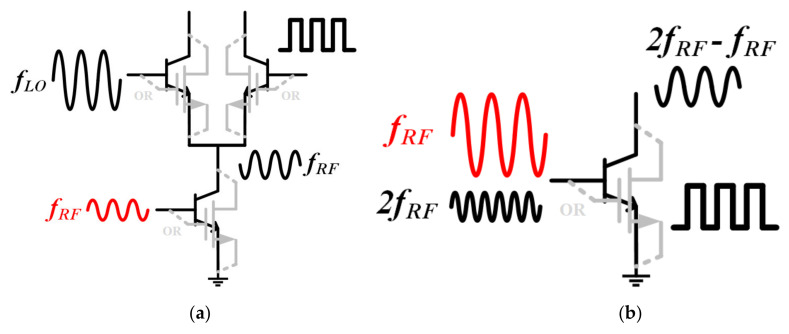
Circuit behaviors under “reverse uplift”. (**a**) Traditional mode, (**b**) switching mode.

**Figure 4 sensors-22-03802-f004:**
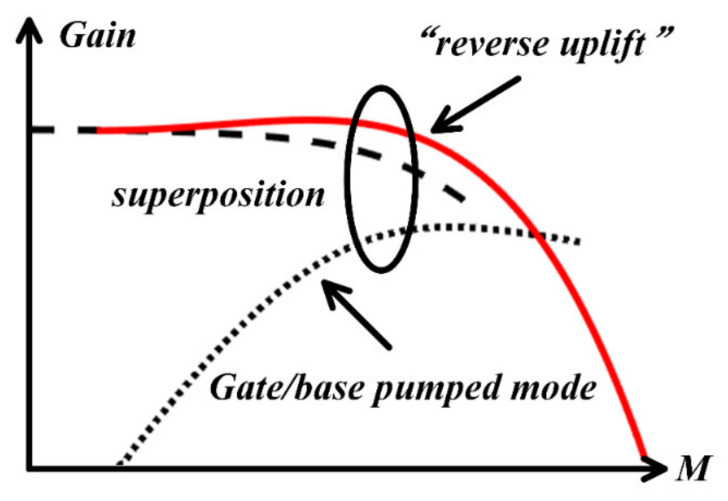
Superposition of the two modes.

**Figure 5 sensors-22-03802-f005:**
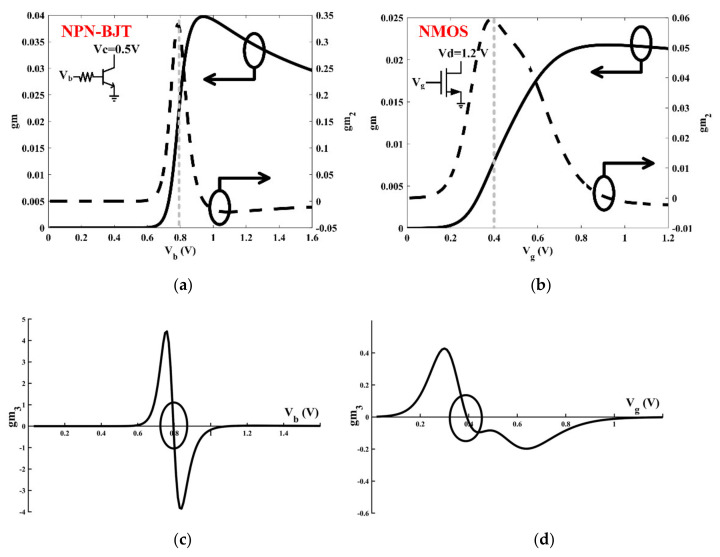
Simulated gm and gm_2_ versus V_b_ (V_g_). (**a**) Negative–positive–negative BJT (NPN-BJT), (**b**) N-channel metal-oxide-semiconductor (NMOS) transistor. Simulated gm_3_ versus V_b_ (V_g_). (**c**) NPN-BJT, (**d**) NMOS transistor.

**Figure 6 sensors-22-03802-f006:**
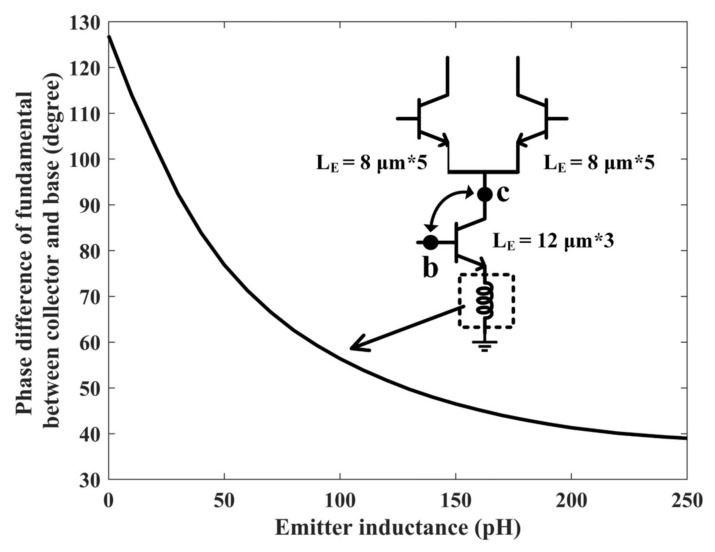
Phase difference of fundamental between collector and base in degree.

**Figure 7 sensors-22-03802-f007:**
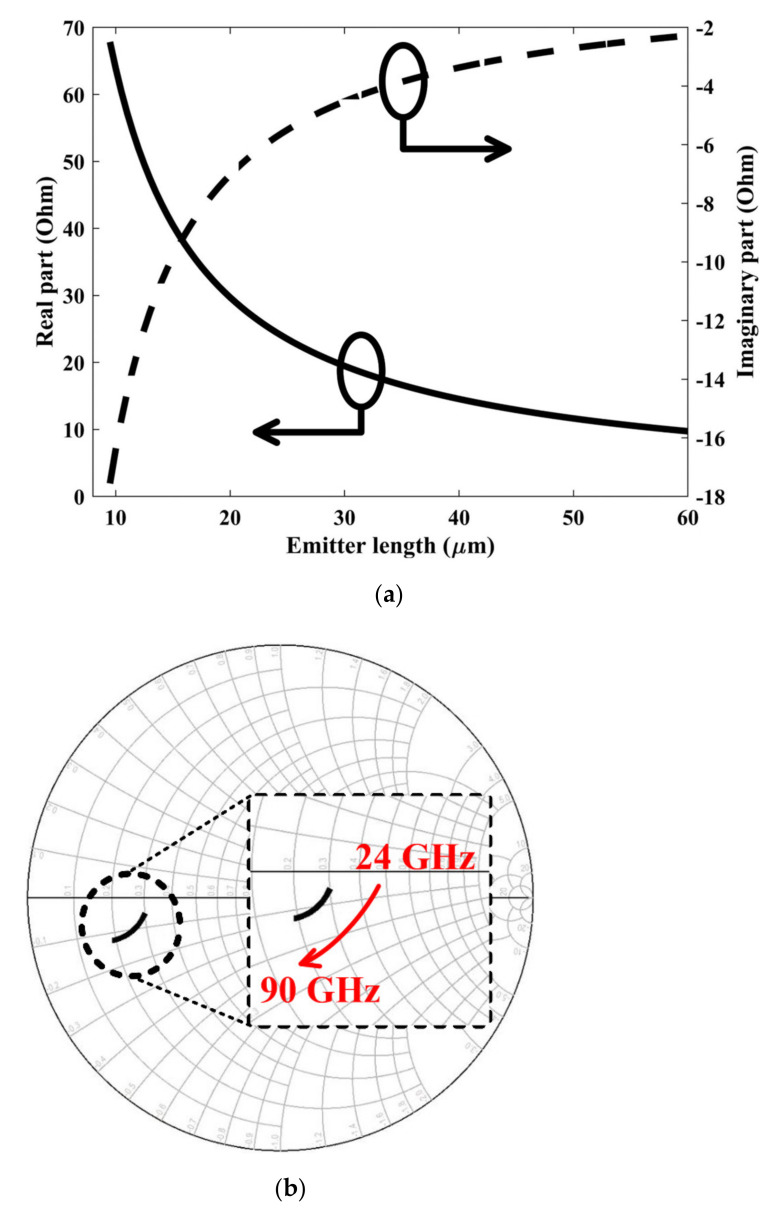
(**a**) Simulated real part and imaginary part of the input impedance of the switching core versus emitter length, and (**b**) the input impedance of the switching core versus RF frequency in Smith chart.

**Figure 8 sensors-22-03802-f008:**
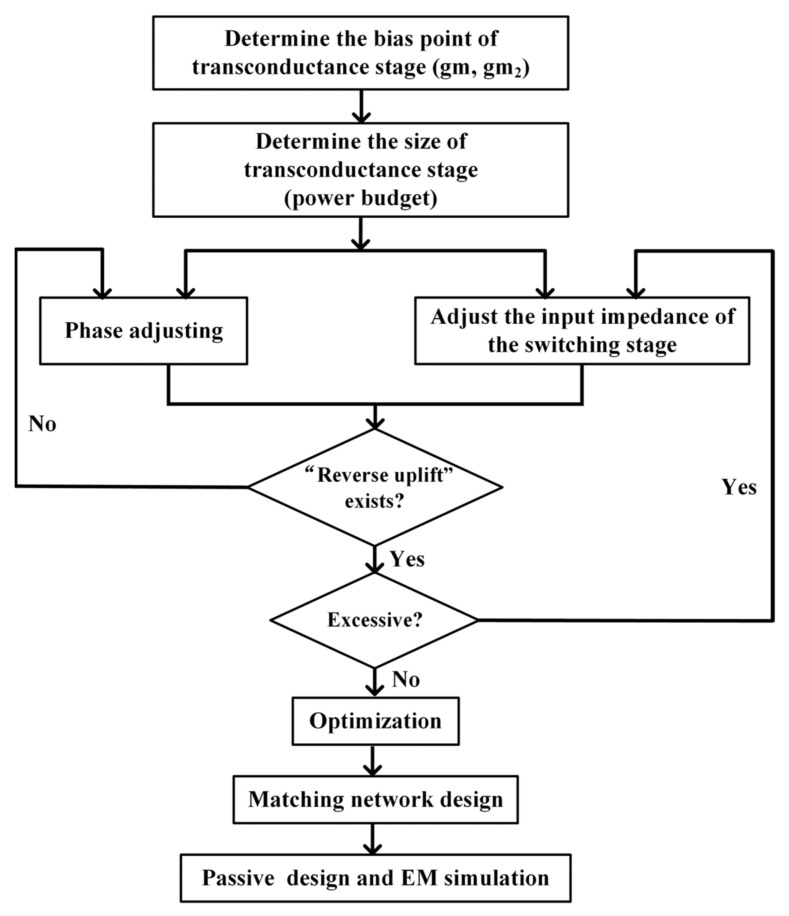
Design process of “reverse uplift”.

**Figure 9 sensors-22-03802-f009:**
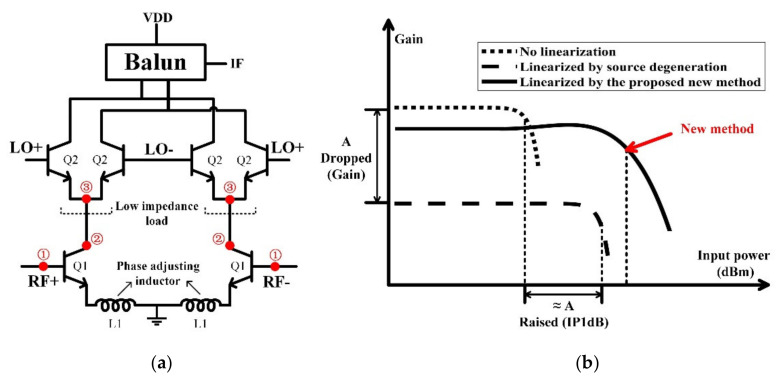
(**a**) The proposed mixer. (**a**) Schematic of the main circuit, (**b**) comparison of the proposed method and source degeneration.

**Figure 10 sensors-22-03802-f010:**
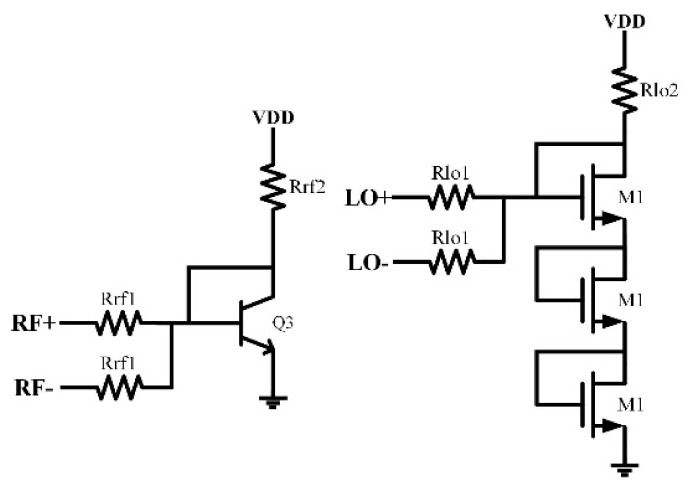
Schematic of the bias circuits.

**Figure 11 sensors-22-03802-f011:**
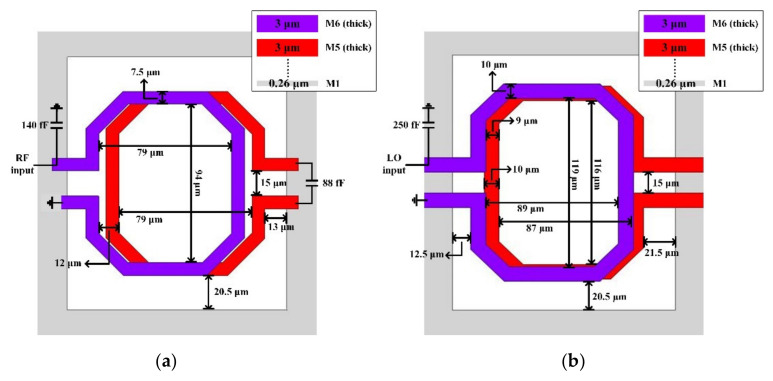
Transformer-based matching networks and detailed parameters, (**a**) RF port, and (**b**) LO port.

**Figure 12 sensors-22-03802-f012:**
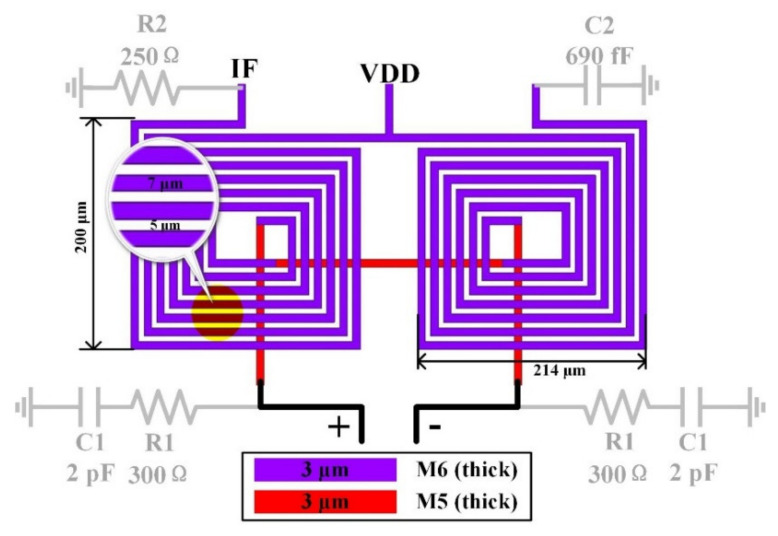
Marchand-balun-based matching network and detailed parameters of IF output port.

**Figure 13 sensors-22-03802-f013:**
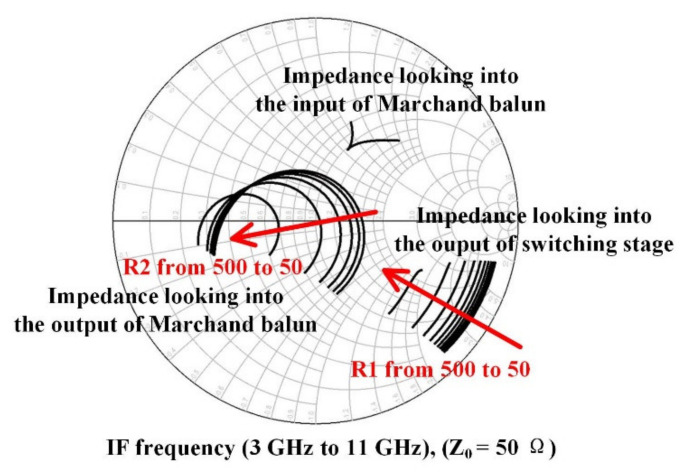
The impedance changes with R1 and R2 at the IF.

**Figure 14 sensors-22-03802-f014:**
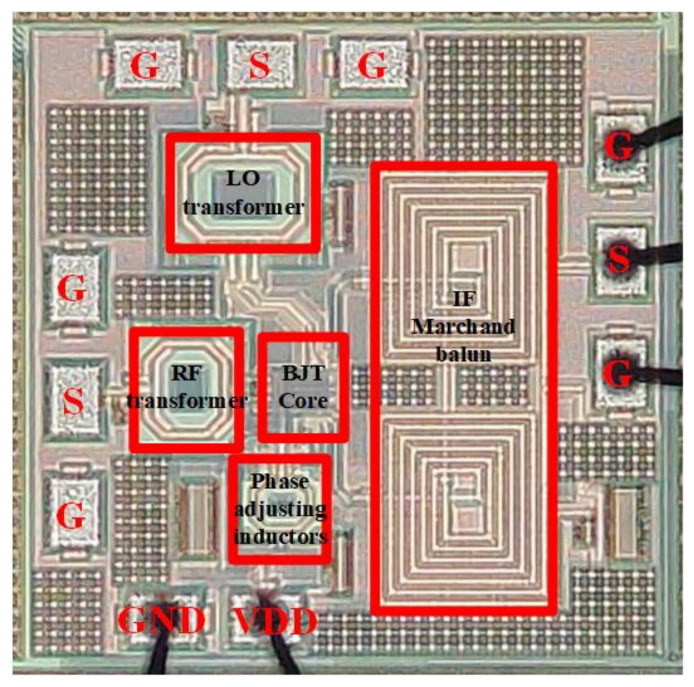
Chip photograph of the proposed mixer with a chip size of 0.69 × 0.69 mm^2^ including pads.

**Figure 15 sensors-22-03802-f015:**
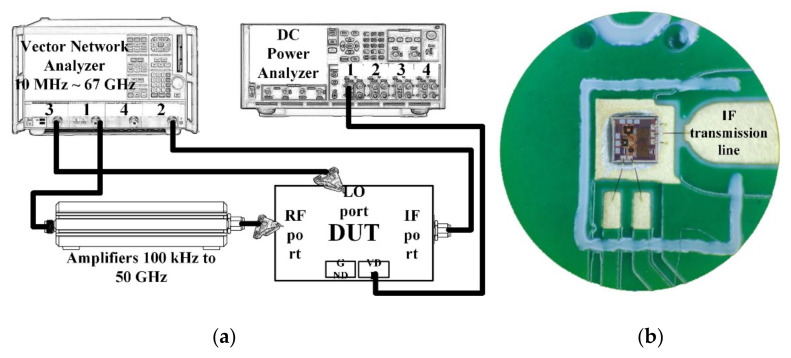
(**a**) Block diagrams of measurement setups, and (**b**) micrograph of the PCB board for bonding.

**Figure 16 sensors-22-03802-f016:**
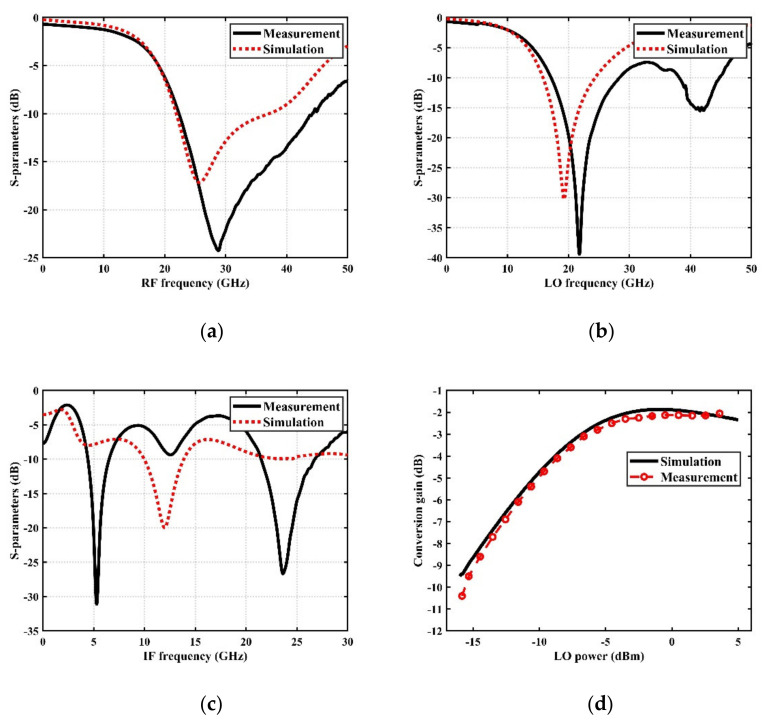
Simulated and measured return losses of (**a**) RF port (S_11_), (**b**) LO port (S_22_) and (**c**) IF port (S_33_). (**d**) Simulated and measured CG versus LO power while RF frequency and LO frequency are 27 GHz and 20 GHz, respectively.

**Figure 17 sensors-22-03802-f017:**
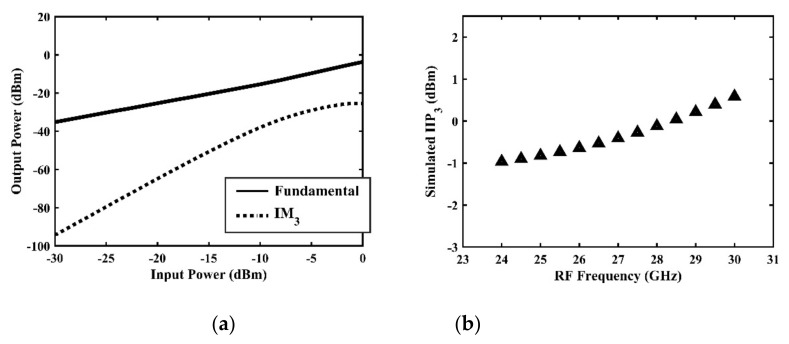
(**a**) Simulated curves of fundamental and 3rd order product, (**b**) simulated IIP_3_ versus RF power.

**Table 1 sensors-22-03802-t001:** Values of some main components.

Q1	L_E_ = 12 μm*3	Rrf1	200 ohms
Q2	L_E_ = 8 μm*5	Rrf2	200 ohms
Q3	L_E_ = 12 μm*3	Rlo1	20 ohms
M1	W = 40 μm, L = 130 nm	Rlo2	200 ohms
L1	80 pH	VDD	1.6 V

**Table 2 sensors-22-03802-t002:** Simulated and measured CG versus RF frequency under different LO frequencies.

LO Frequency	16 GHz	17 GHz	18 GHz
**Simulated and measured CG versus RF frequency**	** 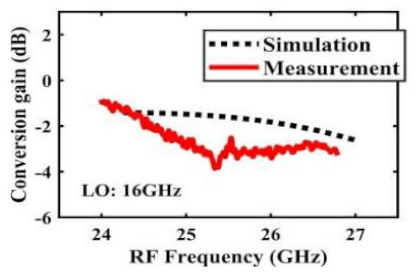 **	** 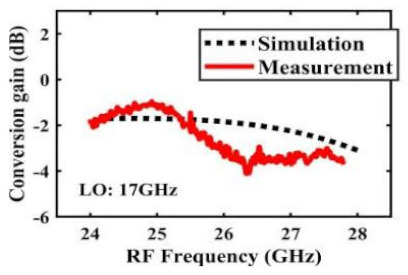 **	** 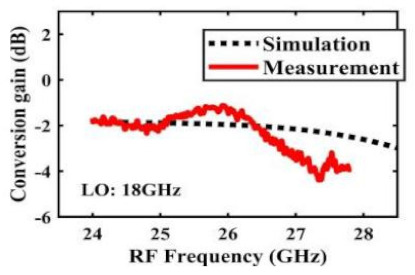 **
**LO Frequency**	**19 GHz**	**20 GHz**	**21 GHz**
**Simulated and measured CG versus RF frequency**	** 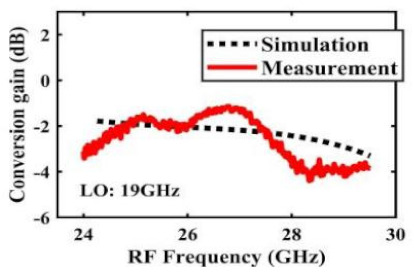 **	** 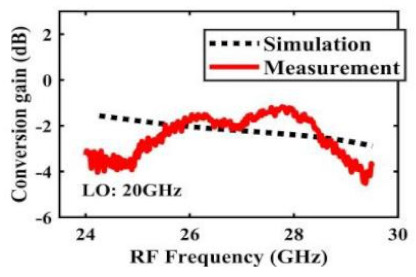 **	** 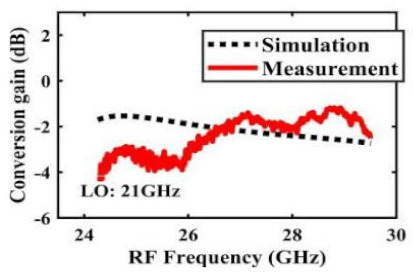 **
**LO Frequency**	**22 GHz**	**23 GHz**	**24 GHz**
**Simulated and measured CG versus RF frequency**	** 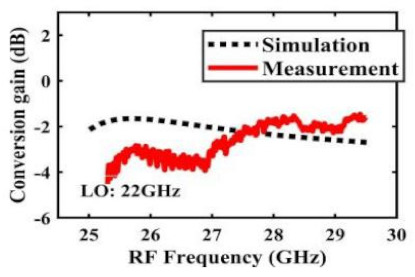 **	** 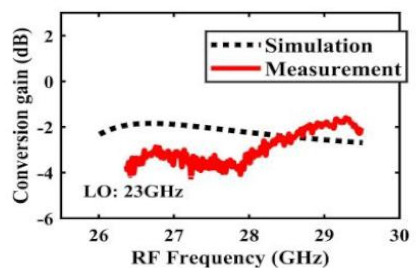 **	** 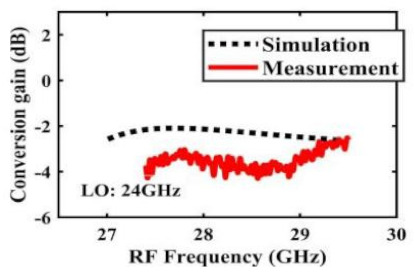 **

**Table 3 sensors-22-03802-t003:** Simulated and measured CG versus input power under nine different RF/LO/IF frequencies.

Frequency	RF: 24 GHz; LO: 17 GHz; IF: 7 GHz	RF: 24 GHz; LO: 18 GHz; IF: 6 GHz	RF: 24 GHz; LO: 19 GHz; IF: 5 GHz
**Simulated and measured CG versus RF power**	** 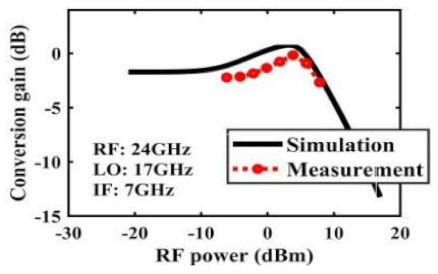 **	** 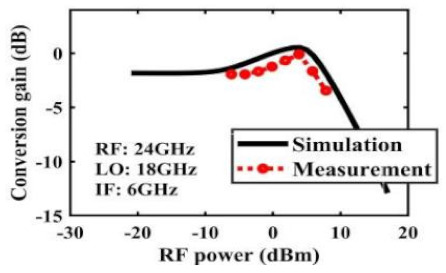 **	** 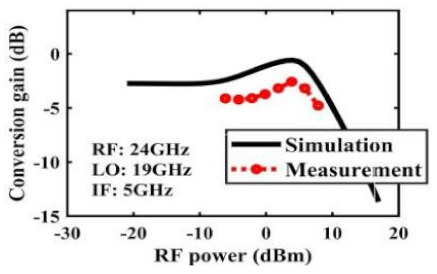 **
**Frequency**	**RF: 24 GHz; LO: 20 GHz; IF: 4 GHz**	**RF: 27 GHz; LO: 16 GHz; IF: 11 GHz**	**RF: 20 GHz; LO: 21 GHz; IF: 9 GHz**
**Simulated and measured CG versus RF power**	** 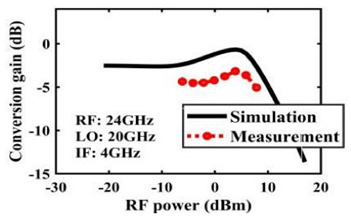 **	** 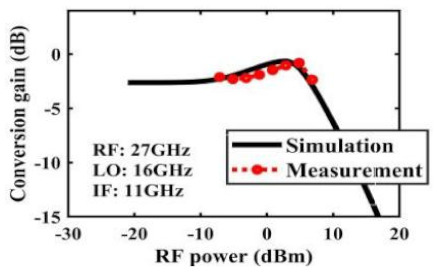 **	** 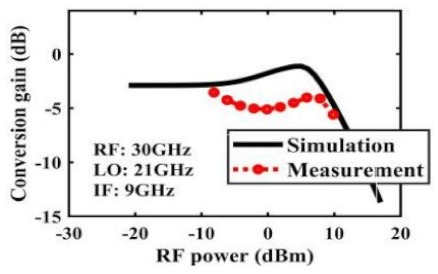 **
**Frequency**	**RF: 30 GHz; LO: 22 GHz; IF: 8 GHz**	**RF: 30 GHz; LO: 23 GHz; IF: 7 GHz**	**RF: 30 GHz; LO: 24 GHz; IF: 6 GHz**
**Simulated and measured CG versus RF power**	** 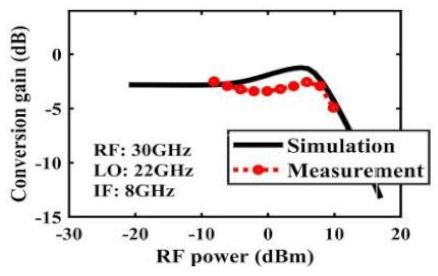 **	** 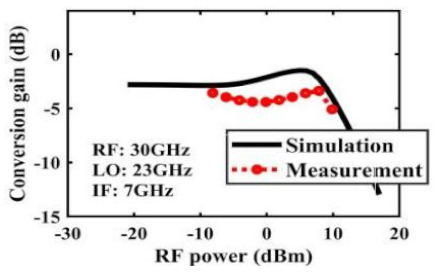 **	** 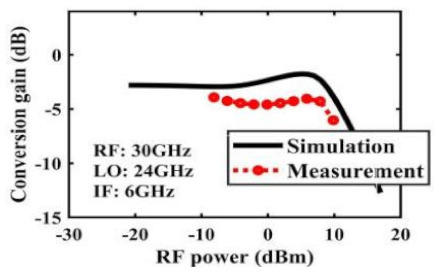 **

**Table 4 sensors-22-03802-t004:** Simulated and measured IP1dB versus RF frequency under different LO frequencies.

LO Frequency	16 GHz	17 GHz	18 GHz
**Simulated and measured IP1dB versus RF frequency**	** 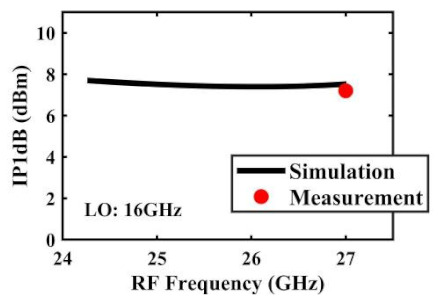 **	** 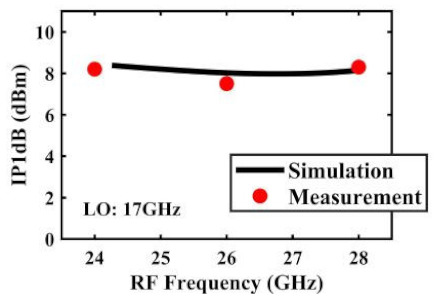 **	** 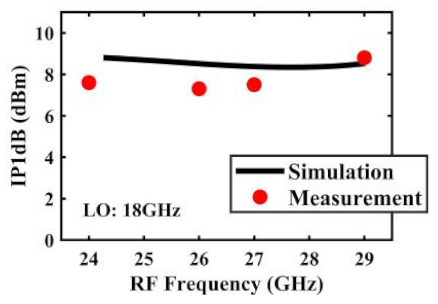 **
**LO Frequency**	**19 GHz**	**20 GHz**	**21 GHz**
**Simulated and measured IP1dB versus RF frequency**	** 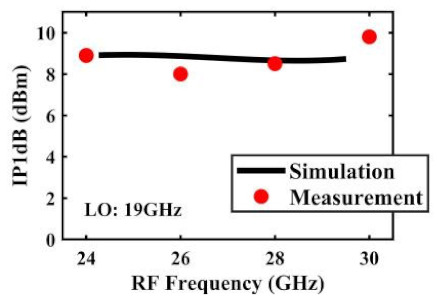 **	** 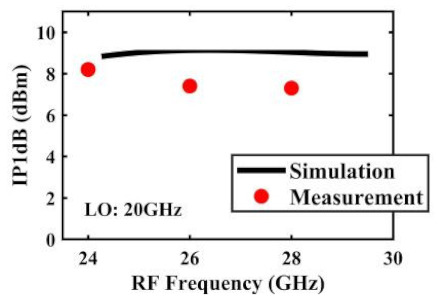 **	** 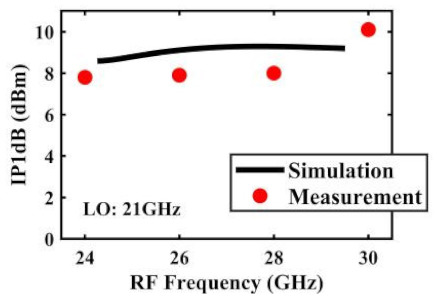 **
**LO Frequency**	**22 GHz**	**23 GHz**	**24 GHz**
**Simulated and measured IP1dB versus RF frequency**	** 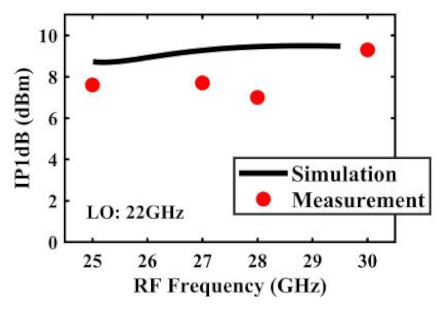 **	** 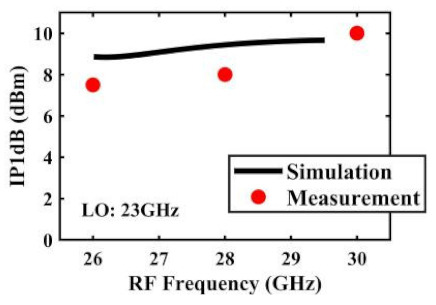 **	** 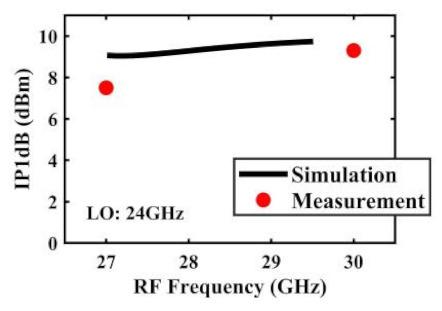 **

**Table 5 sensors-22-03802-t005:** Performance comparison.

Ref.	Process	RF Freq. (GHz)	IF Bandwidth (GHz)	LO Power (dBm)	CG (dB)	IP_1dB_ (dBm)	OP_1dB_ (dBm)	SSB NF (dB)	LO-RF Isolation (dB)	P_DC_ (mW)	Chip Area (mm^2^)	FOM	FOM_IF_
[4]	45 nm CMOS SOI	23~33	4	2	−3.5	−0.2	−4.7	15.3	31	24	0.78	12.25	18.27
[6]	180 nm CMOS	23~25	N/A	5	−4.5 ± 0.6	−4.9	−11	N/A	N/A	16	0.72	6.11	N/A
[16]	90 nm CMOS	35~83	N/A	1	−1 ± 1.5	0	−2	N/A	>30	6.5	0.54	21.12	N/A
[18]	65 nm CMOS	22.5~28.5	1.5	3	17.2 ± 0.2	N/A	N/A	11.2~19.4 (DSB)	>48	7.1	0.88	N/A	N/A
[19]	180 nm SiGe BiCMOS	2~67	0.59	0	2.5 ± 1.4 (20~67 GHz)	−7.4 (40 GHz)	−5.4 (40 GHz)	N/A	>10	17.5	0.42	11.67	9.38
[20]	130 nm SiGe BiCMOS	5~95	4	1	5.5 ± 2.5 (5~90 GHz)	−10 (20 GHz)	−5.5 (20 GHz)	12~6 (Sim.)	50	130	1.2	1.86	7.88
[21]	90 nm CMOS	20~50	2.6	0	0 ± 2	−1	−2	16	>46	6	0.49	22.22	26.37
[22]	90 nm CMOS	5~65	2.5	0	5 ± 1.5	−3	+2	N/A	>30	4.2	0.14	24.02	28
[23]	130 nm CMOS	5~45	5	8	−12.1 ± 1.1	+5.4	−6.9	7.6~10.2	33~47	1.4	0.66	20.29	27.28
This work	130 nm SiGe BiCMOS	24~30	8 (3~11 GHz)	2	−2.3 ± 1.5 (LO:16 GHz) −2.5 ± 1.5 (LO:17 GHz) −2.7 ± 1.6 (LO:18 GHz) −2.7 ± 1.6 (LO:19 GHz) −2.7 ± 1.6 (LO:20 GHz) −2.8 ± 1.6 (LO:21 GHz −3.0 ± 1.5 (LO:22 GHz) −2.9 ± 1.3 (LO:23 GHz) −3.4 ± 0.9 (LO:24 GHz)	+7.2~+10.1	+5.4	9.7~12.1 (Sim.)	>34.5	19.8	0.48	24.73	33.76

## Data Availability

Not applicable.

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
