# Peer review of "A 24-to-30 GHz Ultra-High-Linearity Down-Conversion Mixer for 5G Applications Using a New Linearization Method"

_sensors, 2022, doi:10.3390/s22103802_

Round 1

Reviewer 1 Report

Dear authors;

Your paper entitled “A 24-to-30 GHz Ultra-high Linearity Downconversion Mixer for 5G Applications Using a New Linearization Method” I have several comments that I wish to be useful for you:

1- The abstract needs more interest and rewriting some paragraphs.

2- There are still some aspects that can be improved (for grammar and punctuations). Improve the technical writing of your paper, where there are several grammatical errors and spelling I think they need to be checked out.

3- The conclusion needs more efforts to elaborate the achieved results with respect to the future work,

4- The practical part is very important,

5- Future work is an important part of the conclusion.

I loved this work and I feel it is very good. I hope these comments would help you improve this work after a major revision.

Regards

Author Response

Immediately after receiving your comments, we made revisions and uploaded a new manuscript overnight, thank you very much for your review.

Reviewer 2 Report

Please see the minor comments below which will help to improve your work.

================================
- In the abstract section, the author tried to explain the methods used and the results clearly. However, the author should include two points to make your abstract more interesting,  such as 
  a problem statement ( problem that make you propose solutions)before explaining about methods, and also add contribution at the end of the abstract.

- In the introduction section, the style of listing "24.5/28/37/39/43 GHz" is recommended change to "24.5, 28, 37, 39, and 43 GHz".

- If there are many continued citations, it should be write in single bracket. For example, the multiple.....[13-15]...... sacrificing CG.
  And if citations are not continued, please use "," to separate each citation in a single bracket. For example, [1,4].

- Please check again the equation 1, you should use a variable for example "n" to represent the increasing 1,2,3,...n.

- To prevent misunderstanding the meaning of V/I conversion and V/I converter, please add what V/I conversion and V/I  refer to, and their meaning.

- Please check the whole manuscript again and be sure to give full meaning before using the abbreviation. For example output 1 dB compression point (OP1dB).

- Using "S-parameter" is recommended instead of "S parameter".

- What software does the program use to perform the simulation?

- What is the serial number of the components used, serial numbers of transistors, etc?

- There are many typos in the description of Figure 17(a)~(i), so please revise it. In addition, if possible please use a table for the presentation.

- According to the result report in Table 2 show that your work achieved higher IP1dB and higher OP1 than any review works.
 Therefore, please give more explanation about what happens and the importance when your IP1dB and OP1 are high.

- The return loss is also important and needs to compare with previous works, please add return loss in Table 2 Performance comparison.

- Experiments were conducted under a lot of conditions, but no discussion of why the results turned out that way and discussion about figure results (Fig16-20).

- The conclusion is too short, please add more descriptions.

Author Response

(The authors gave the same response as above.)

Reviewer 3 Report

This manuscript proposed a new circuit technique to improve the linearity of a mixer. Using this technique, P1dB is significantly improved.  However, for the IM3 performance, there are only simulation results.  From Fig. 20(b),  it is observed that the IIP3 (around -1 dBm) is much worse than the IP1dB in Figs. 19.  Since IIP3 is a key factor for the linearity, it is suggested to explain the reason why IIP3 is not improved as the P1dB.

Author Response

(The authors gave the same response as above.)

Reviewer 4 Report

- The proposal is appealing and interesting, and the method deserves some consideration. Moreover, the paper is almost well written and well organized.

- The text, in general, reads well, but a quick grammatical revision could improve it further.

- The paper adequately puts the progress it reports in the context of previous work, representative referencing and first discussion.

- The authors could highlight better the new scientific contribution.

Author Response

(The authors gave the same response as above.)

Round 2

Reviewer 1 Report

The revised version is good

Author Response

Thanks for your review.

Reviewer 2 Report

Thank you for your effort to revise and  explain including reviewer's comments and self revision to improve work. 
However, there is have minor error after revised. Please see the minor comments below which will help to improve your work.

================================
- Please rearrange many equations at line 468, 469, and 470. Those equations should write separate line (one equation one line) with equation label number at the end.
  For example: FOM = 10log()             (15).

- There are several abbreviations errors: please revise them as the following.
   - NPN-BJT (Negative-Positive-Negative BJT)  to Negative-Positive-Negative BJT (NPN-BJT)
   - NMOS (N-Channel Metal-Oxide-Semiconductor) to N-Channel Metal-Oxide-Semiconductor(NMOS)
   - RF (Radio Frequency) to Radio Frequency(RF)

- Typo error at line 412, please revise "When" to lowercase letters "when".

- Referring to Table 2. Performance comparison, the authors compared their work to previous studies [8], [18],[19], [20], [21],[22],[16], [23], and [10] without write them down first in related work.
  Therefore, I recommend add description of those works in related work section before referring in Table 2 of manuscript.

- List references of those works should be in continue number as [8],[10], [16], [18], [19], [20], [21],[22], and [23].

Author Response

Thanks for your review. We've made changes to each of your suggestions, thank you very much.
